# “It’s Different for Girls!” The Role of Anxiety, Physiological Arousal, and Subject Preferences in Primary School Children’s Math and Mental Rotation Performance

**DOI:** 10.3390/bs14090809

**Published:** 2024-09-12

**Authors:** Michelle Lennon-Maslin, Claudia Michaela Quaiser-Pohl

**Affiliations:** Department of Developmental Psychology and Psychological Assessment, Faculty of Educational Sciences, Institute of Psychology, University of Koblenz, 56070 Koblenz, Germany; quaiser@uni-koblenz.de

**Keywords:** math anxiety, physiological arousal, math performance, mental rotation skills, gender differences, primary education

## Abstract

(1) Background: This study examines the role of subjective anxiety (mathematics and spatial anxiety), along with physiological responses, in mathematics or math and mental rotation performance in 131 German primary school students (65 girls, 66 boys; *Mean age* = 8.73 years). (2) Method: Students’ preference for math vs. German and their subjective anxiety were assessed using standardized questionnaires. Emotional reactivity was measured using the Galvanic Skin Response (GSR). Math performance was evaluated via percentage scored and completion times on number line estimation, word problems, and missing terms tasks. Spatial skills were assessed using a novel mental rotation task (nMRT) incorporating gender-congruent and -neutral stimuli. (3) Results: Girls outperformed boys on percentage scored on the math task but took longer to complete this. No gender differences were found in performance on the nMRT. Girls demonstrated higher math anxiety and were less likely to prefer math over German. Math anxiety predicted math scores and accuracy on the nMRT while gender predicted math performance and mental rotation response time. Subject preference was associated with longer completion times and emotional reactivity with longer response times. Girls’ preference for math and lower emotional reactivity was linked to shorter completion times, while lower math anxiety predicted higher scores. In contrast, these factors did not affect boys’ math performance. Additionally, subjective anxiety, emotional reactivity, or subject preference did not impact spatial performance for either gender. (4) Conclusions: Supporting mathematical self-efficacy and emotional regulation, especially in girls, is crucial for enhancing STEM outcomes in primary education. Gender-fair assessment in mental rotation reveals equitable spatial performance and reduces the impact of anxiety.

## 1. Introduction

The recent social media phenomenon, “Girl Math”, which began as a light-hearted viral joke about women’s alleged shopping and bargaining habits, has also reignited a persistent stereotype that women have a negative relationship with mathematics [1]. In a recent interview, British mathematician Hannah Fry, who describes herself as a “numberphile”, discussed the low number of women pursuing careers in math-related fields. When asked about the reasons behind this gender disparity, Fry stated that it cannot be attributed to a single cause. Instead, she described it as “death by a thousand cuts” [2]. Fry elaborated that these ‘cuts’ collectively reinforce the notion that women do not belong in the field, highlighting a significant impact of stereotype threat. The article further outlines evidence from research which consistently demonstrates that stress and math anxiety severely affect girls’ focus, undermining their working memory and limiting their ability to perform well in the subject [3,4,5]. Poor female math performance perpetuates the gender stereotype that “girls are just not good at math” a belief which, despite their desire to excel, many girls internalize hindering their success in the field [2]. Therefore, the aim of this study is to investigate the role of emotion, particularly anxiety, in the performance of primary school children on mathematical and spatial tasks. By measuring specific forms of subjective anxiety as well as physiological arousal, this study seeks to unravel the complex relationship between these psychological factors and their influence on performance in early education.

### 1.1. Gender Differences in Mathematics Performance in Primary School Students

Evidence for gender differences in math performance in primary school is mixed and context-dependent. Historically, studies indicate no significant differences between boys and girls in early education settings. Meta-analyses and longitudinal studies have generally found that any observed gender differences are small and often not statistically significant. For instance, Hyde et al. (2008) [6] conducted a meta-analysis in the USA and found gender differences in math performance to be negligible. However, gender differences can vary significantly by country and cultural context. Cross-national studies, such as those by the Program for International Student Assessment (PISA), show varying gender gaps across different educational systems [7]. The 2018 PISA study found that in Germany, the gender gap in math performance among adolescents was wider compared to the OECD average [7]. Research suggests that stereotypes and peer influence contribute to this outcome. Wolff (2021) [8] found that in German secondary schools, stereotypes like “boys are better at math than girls” negatively impacted girls’ self-concept but not boys’, even after controlling for grades and age. Gender stereotypes regarding math ability can emerge as early as primary school, affecting confidence and interest in the subject [9,10]. Additionally, negative cultural stereotypes such as the “STEM-nerd” influence the attractiveness of Science, Technology, Engineering, and Mathematics (STEM) and related careers, often limiting girls’ identification with and choice of related fields [11]. The effects of societal stereotypes and associated thinking can also affect STEM subject preferences as early as primary and secondary school, leading to gender differences in the pursuit of mathematically related studies and STEM careers going forward [12,13,14].

### 1.2. Mental Rotation and the Role of Stereotypes in Its Assessment

Mental rotation (MR), the ability to mentally manipulate two- and three-dimensional objects, is a critical spatial skill extensively examined in psychology and education [15,16]. MR is typically assessed using psychometric instruments such as the mental rotation test (MRT), which usually consists of items with cubical stimuli (Vandenberg and Kuse, 1978 [17]). These tasks consistently show large and reliable performance differences favoring males [18]. Traditionally, MR tasks have shown a male performance advantage, potentially due to boys’ greater engagement in spatially oriented play [19,20]. However, recent studies indicate that the design of MR tasks, including the use of gender-stereotyped or -neutral stimuli, can influence these gender differences by either reinforcing or mitigating stereotype threat [16,21]. Findings indicate that the familiarity or gender congruence of stimulus objects significantly influences performance differences. There is a positive correlation between the stereotyped content of the stimuli and children’s MR performance [22,23,24].

### 1.3. The Association between Mental Rotation and Mathematical Performance

Numerous studies confirm the association between spatial and mathematical abilities [25,26,27]. Mental rotation (MR), like other spatial reasoning skills, can be improved through targeted practice and training, which often translates to better mathematical performance [28,29]. MR skills are linked to advanced mathematical skills and are strong predictors of mathematical competence in young children [30]. Moreover, MR performance is a significant indicator of academic success and career choices in STEM fields [31]. Similar to mathematical ability, gender differences in MR performance are influenced by stereotype threat, self-concept, and emotional arousal. These factors tend to negatively affect women’s task performance due to a fear of confirming negative stereotypes about female weaknesses in visuo-spatial cognition [32,33,34].

Gender differences in performance on math and spatial tasks may be particularly evident in processing speed. Limited working memory capacity constrains speed, impacting completion, reaction, and response times for cognitive tasks like math and MR. Girls may be more sensitive to stress from time pressure on a math or spatial test or fear making mistakes, which diverts their cognitive resources, slowing the mental operations needed for rapid and accurate responses on timed tasks [34,35]. Boys, on the other hand, tend to demonstrate higher math self-concept and confidence in their mathematical ability [13]. They often overestimate their aptitude and are more influenced by peer dynamics, with social comparisons shaping their confidence in math [36]. Additionally, boys’ tendency to prioritize speed over accuracy which can result in shorter processing times on cognitive tasks, potentially sacrificing thoroughness and leading to errors [37,38]. There is evidence from research that girls’ performance on mathematical and spatial tasks is slower than boys’, which in turn leads to a disadvantage in their overall performance in these areas [39,40]. This may be due to a variety of factors such as the effects of stereotype threat, higher anxiety levels during the tasks, and diverse processing strategies [34,41,42,43].

### 1.4. Interplay of Psychological Factors and Their Role in Mathematical and Spatial Performance

Mathematics performance is a fundamental component of academic achievement and cognitive development in primary school children [44]. Understanding mathematical concepts is crucial for success in various educational domains, including STEM fields [45,46]. However, individual differences in math performance are influenced by factors beyond cognitive abilities, such as self-concept and emotional arousal [47,48].

Mathematics anxiety, characterized by fear or apprehension during mathematical tasks, significantly hinders learning and achievement by depleting working memory and leading to poorer cognitive performance [49,50]. High math anxiety often results in reduced motivation and avoidance behaviors, contributing to underperformance [42,51,52]. Compared to boys, girls’ appear to experience more negative effects of math anxiety on performance [53,54]. Similarly, spatial anxiety, which involves discomfort with spatial reasoning tasks, is linked to deficits in spatial abilities, particularly in girls and women. It can disrupt performance on mental rotation tests, especially when gender stereotypes are at play [41,55,56]. Hence, anxiety in both math and spatial tasks can hinder children’s cognitive performance.

Recent studies highlight the interconnectedness of psychological factors in gender differences in mathematical and spatial performance. Spatial anxiety and spatial self-efficacy have been found to mediate gender differences in mental rotation performance, particularly in challenging tasks [57]. Additionally, mental rotation performance and perceived competence mediate the relationship between gender and math anxiety in younger populations [58]. Research also links statistics anxiety with spatial processing and performance, and recent findings suggest that lower math self-concept in girls may affect accuracy in mental rotation, with spatial anxiety playing a mediating role [13,59]. These studies highlight the interconnectedness of mathematical and spatial skills and associated anxieties.

### 1.5. Physiological Markers of Emotion Regulation and Their Role in Spatial and Mathematical Ability

Arousal, representing the level of central nervous system activity, plays a critical role in regulating the human stress response [60]. Transactional models of stress suggest that various events, tasks, and conditions, as well as the individual’s assessment of these, trigger physiological, cognitive, behavioral, and emotional responses [61]. Measuring physiological markers, such as systolic blood pressure and electrodermal activity (EDA), offers valuable insights into emotional reactivity and cognitive engagement during learning tasks [62,63].

Galvanic Skin Responses (GSRs) serve as biomarkers of autonomic nervous system (ANS) arousal and are well established for assessing psychophysiological functioning in humans [64,65]. The GSR effectively captures physiological changes associated with emotional expressions such as anxiety and stress in children [66,67]. Wearable devices for GSR measurement facilitate the examination of physiological responses in children engaged in cognitive tasks under real-life conditions [68,69]. Galvanic Skin Responses (GSRs) in children vary depending on the context in which they are measured. These variations are influenced by the emotional and stressful nature of tasks, reflecting differences in emotional arousal and stress responses [67,70]. For example, anxiety linked to math tasks is associated with increased vigilance and heightened amygdala activity, leading to greater physiological reactivity [48,71]. Mental rotation tasks, due to their demand for spatial reasoning, can trigger significant stress and emotional arousal, especially when new to children [41]. In contrast, familiar math tasks practiced regularly in school may reduce anxiety and physiological arousal [72].

The ability to regulate emotions impacts performance on spatial tasks, with gender differences in skin conductance responses observed during mental rotation tasks [73,74]. A pilot study by Lennon-Maslin and colleagues (2023) highlighted that increased physiological arousal, measured by GSR, influenced primary school children’s response times on specific items in a novel mental rotation task [75]. These physiological responses are context-dependent, influenced by factors such as stimulus type, measurement context, emotion regulation strategies, and sociocultural influences [76]. Understanding these variables is crucial for interpreting GSR differences among children in educational settings.

### 1.6. Aims of the Current Study

Despite growing interest in the role of anxiety and physiological arousal in mathematics and spatial performance [48,62], limited research has investigated their combined effects in primary school children. Moreover, only one pilot study to date has examined the role of emotional reactivity measured by the GSR in mental rotation performance in the same target group [75]. The present study, therefore, represents a novel contribution to research in this field by examining the simultaneous impact of math and spatial anxiety, along with physiological arousal measured through skin conductance, on both mathematical performance and mental rotation skills in primary school children. By utilizing standardized measures for anxiety and physiological assessment, this study aims to provide an unprecedented, comprehensive understanding of the interplay between emotional reactivity and anxiety-related psychological factors in shaping early cognitive performance. This integrated approach not only addresses a critical gap in the literature but also offers valuable insights which could inform educational interventions designed to enhance learning outcomes in young children.

### 1.7. Research Questions

Based on an extensive review of the literature, our study set out to explore some key questions about gender differences in performance on math and spatial ability among primary school children. Specifically, we sought to determine if there are noticeable differences between boys and girls in their performance on mathematical and spatial tasks and their association with task-related subjective anxiety, emotional reactivity, and their preferences for school subjects.

Our first research question seeks to answer whether there are gender differences in the performance of our sample on cognitive tasks, specifically in math and mental rotation, in subjective anxiety (math and spatial anxiety), and in the preference for math vs. German. This question addresses a critical gap in the literature, as previous studies have shown mixed and context-dependent findings regarding gender differences in these areas [6,7,8]. By examining these variables in primary school children within a specific cultural context, our study aims to clarify these inconsistencies.

Our second research question delves into the interplay between several factors—gender, subject preference, emotional reactivity (as measured by the Galvanic Skin Response or GSR), and subjective anxiety (both math and spatial anxiety)—and how they collectively influence performance on cognitive tasks, specifically in math and mental rotation. While previous research has extensively explored these variables individually, few studies have examined how they interact to influence cognitive performance in young children, particularly in a primary school setting [42,48,77]. This research question seeks to address this gap by providing a more holistic view of how these factors combine to affect performance. 

### 1.8. Hypotheses

**H1.** 
*We hypothesized that there would be significant gender differences in both the percentage scored and the completion times on the math task. We further hypothesized that girls would outperform boys in terms of the percentage scored but would take longer to complete the task.*


Rationale: This hypothesis explores the idea that girls often perform just as well or better than boys in math accuracy but often at the cost of speed, potentially due to increased anxiety and diverse problem-solving strategies [40,43].

**H2.** 
*We expected no significant gender differences in terms of accuracy and response time on the mental rotation task.*


Rationale: Recent research challenges traditional beliefs about gender differences in spatial abilities, particularly when tasks are designed to minimize stereotype threat [22,23,24,75]. This hypothesis also tests whether these findings hold in a primary school context.

**H3.** 
*We anticipated that girls would report higher levels of subjective anxiety (both math and spatial) compared to boys.*


Rationale: This expectation aligns with the existing literature, which frequently documents higher anxiety levels among females in math-related contexts [53,54] and also on spatial tasks [13,41,55]. Studies exploring gender differences in spatial anxiety in primary school children are particularly limited, as are studies examining its effects on math performance [13,78].

**H4.** 
*We hypothesized that girls in primary education would be less likely than boys to choose math as their preferred school subject over German.*


Rationale: Although some studies have examined how stereotypical thinking can lead to gender differences in the preference for mathematics and STEM as a subject at secondary school and beyond, few studies have examined this in the primary school context [12,13,14]. This hypothesis addresses a gap in understanding how early subject preferences play a role in performance on math and spatial tasks in primary education. This may have implications for the pursuit of STEM studies and careers, a field in which gender disparities are most pronounced [79,80].

**H5.** 
*We hypothesized that there would be a significant association between gender, subject preference, and performance on the math task when controlling for emotional reactivity and math and spatial anxiety. For girls, we expected that a preference for math, lower emotional reactivity, and lower math and spatial anxiety would predict higher percentages scored and shorter completion times on the math task compared to boys.*


Rationale: This hypothesis aims to identify potential protective factors, such as subject preference, reduced subjective anxiety and physiological arousal that might mitigate potential negative impacts on performance [13,59,81]. This area remains underexplored in the literature.

**H6.** 
*We hypothesized that there would be a significant association between gender, subject preference, and performance on the mental rotation task when controlling for math and spatial anxiety and emotional reactivity. However, we expected no significant gender differences in how these factors affect performance on the mental rotation task, that is, the effects would be similar for both girls and boys.*


Rationale: Anxiety, emotional reactivity, and physiological arousal are known to affect performance on complex cognitive tasks such as mental manipulation which may trigger stereotype effects known to be associated with spatial ability [48,67,73]. This hypothesis examines whether commonly observed gender differences in emotional and physiological factors persist on a mental rotation task consisting of gender-congruent and -neutral stimuli.

## 2. Materials and Methods

A total of one hundred and thirty-one students were recruited from primary schools in the state of Rhineland Palatinate in Germany (*N* = 131). All of the students belonged to second-, third-, and fourth-grade classes, respectively, and were enrolled in five local primary schools in Rhineland Palatinate, Germany. Schools were situated in both urban and rural settings. Although all of the children were fluent German speakers, some were also bilingual and had diverse ethnic backgrounds. Unfortunately, specific data regarding ethnicity and socio-economic status was not collected from parents and guardians. These observations were made by the researchers and through information volunteered by children prior to testing.

All students completed the math task, the mental rotation task, and various questionnaires. Included in this sample were 66 students who identified as boys and 65 as girls. There were no students who identified as non-binary in this sample. The average age of the students was *M* = 8.73 (*SD* = 1.02) years old. Mean grade level distribution by gender is shown in the bar chart in Table 1.

### 2.1. Skin Conductance

Shimmer3 GSR+ Unit^®^ (https://shimmersensing.com/product/consensyspro-software/, accessed on 1 January 2024) was used to measure galvanic skin response (GSR). These devices were synchronized with ConsensysBasic^®^ multi-sensor management software (https://shimmersensing.com/product/consensyspro-software/, accessed on 1 January 2024) where they were calibrated for recording and a sampling rate of 1 Hz was set. A baseline recording of 2 min per participant was planned in order to compare this with GSR during the math task and during the MRT.

Shimmer devices were attached to the index and middle fingers of the non-dominant hand. This placement allows for optimal signal detection while minimizing interference with the tasks performed using their dominant hand. Each child was asked to try and keep the monitored hand as still as possible throughout the measurement period. This was to ensure consistent and accurate readings by minimizing movement artifacts. The GSR measurement began with a 2 min baseline recording. During this period, the children were asked to sit quietly and relax. This baseline data served as a control to compare physiological arousal during the subsequent tasks. After the baseline measurement, the students performed a series of math problems. The GSR device continued to record physiological responses throughout this task. Following the completion of the math task, students proceeded to the mental rotation task. The GSR measurement continued during this period to capture physiological responses associated with this cognitive activity.

Once the tasks were completed, the devices were redocked to stop the recording. This was completed before administering the self-report questionnaires to ensure that the GSR data were isolated to the periods of baseline, math, and mental rotation tasks. The measurements were conducted in a controlled environment, specifically a separate classroom with adequate lighting and individual seating arrangements. This setup aimed to minimize external distractions and create a comfortable setting for the children to perform the tasks. The room temperature and other ambient conditions were kept consistent to avoid any environmental factors that might influence physiological responses. This careful administration and controlled environment ensured the reliability and validity of the GSR measurements, allowing for an accurate assessment of the children’s physiological arousal in response to the cognitive tasks.

Skin conductivity is measured in units referred to as microsiemens (S). “Micro” is a prefix meaning millionths, so 1 microsiemen (1 S) is a unit of time in the International System of Units (SI) equal to one millionth of a siemen [82].

### 2.2. Math Task

Students were given a paper-and-pencil math task consisting of 9 items of varying difficulty levels: A, B, and C. Each item had 5 possible solutions, with only one correct answer. A-level tasks were the easiest, worth 3 points each, B-level tasks were more difficult, worth 4 points each, and C-level tasks were the most difficult, worth 5 points each. Second- and third-grade students had to solve three A-level, four B-level, and two C-level problems and could score a maximum of 35 points, while fourth-grade students could score 36 points due to an additional C-level problem. The problems were drawn from the Mathematical Kangaroo Competition an international math challenge in over 77 countries, the purpose of which is to show students that math can be interesting, beneficial, and even fun [83]. The Kangaroo math problems have been translated into many languages; therefore, for our task, we drew the problems in the German language from the “Mathematikwettbewerb Känguru” website [84]. To ensure unfamiliarity, problems from past competitions were chosen. Our math task included three types of math problems: number line estimation, word problem representation, and missing terms. Math performance was measured based on the percentage scored and the completion time, that is, the interval from when the task is presented to when the participant finishes solving the problems and submits their answers.

Students received written instructions to work individually, circle or mark answers, and refrain from using a calculator or other electronic aid. Zero points were given for unanswered questions. The task aimed to engage students in math and minimize fatigue. An example of each type of math problem used in the task is listed in Appendix A.

### 2.3. Mental Rotation Task

A novel, computerized mental rotation task (nMRT) based on Vandeberg and Kuse’s original MRT but consisting of more varied stimuli was programmed in PsychoPy^®^ software (www.psychopy.org/download.html, accessed on 6 April 2023) and installed on Microsoft Pro 8 Surface tablets, each with a keyboard and a mouse. The task was programmed to record both the number of correct responses (accuracy) as well as the duration in seconds from when the stimulus is presented to when the participant provides the answer (response time). Items included mental rotation stimuli for younger children, i.e., animals and letters [85], as well as other concrete stimuli and abstract stimuli, all either rotated in-depth or in picture plane. Abstract items consisted of stimuli such as cubes, pellets [22], and polyhedral figures [24]. Other concrete items consisted of male and female gender-stereotyped stimuli [21,23] and gender-neutral stimuli [24]. Examples of some of the items used are shown in Figure 1 and are also appended to this report. 

The MRT was divided into two parts, each with a time limit. Part one (MRT 1) consisted of 6 abstract and 10 concrete items which were rotated in picture plane only. It was limited to 5 min. Part two (MRT 2) contained 6 abstract and 6 concrete items rotated in-depth and was limited to 8 min. Items were presented randomly in each part of the task with one target stimulus on the left and four comparison stimuli on the right. Participants were instructed to identify two out of four stimuli on the right which, although rotated, were identical to that on the left. A reliability analysis of MRT conducted on the data from our sample yielded a Cronbach’s Alpha of *α* = 0.86.

### 2.4. Demographic Data

An online questionnaire was presented to each participant following the MRT to collect data relating to participants’ age and gender. The position of this and the other self-report questionnaires in the procedural sequence, i.e., being placed after the math and mental rotation tasks, aimed to avoid priming or eliciting stereotype threat in male and female participants.

### 2.5. Self-Report Questionnaires

Three further self-report questionnaires were also presented in PsychoPy^®^, the purpose of which was to assess the students’ preference for math vs. German, their spatial anxiety, and their math anxiety. All the questionnaires were presented to students in the German language.

#### 2.5.1. Math versus German Preference Survey

Students were surveyed to indicate their preferred school subject among math, German, or no preference for either subject. Their responses were coded in SPSS for statistical analysis as follows: math = 1; German = 2; no preference = 3. This question is part of the Academic Self-Concept in Primary School Children (ASKG) questionnaire, a German self-report measure of reading, writing, and math self-concept for children [86]. We used only the Math vs. German survey section, which is appended to this article.

#### 2.5.2. Child Spatial Anxiety Questionnaire

The Child Spatial Anxiety Questionnaire (CSAQ) [41] consists of 8 items in which students are asked to rate how anxiety-provoking they find a particular task. All tasks described in the questionnaire require spatial ability and skills. Researchers explained to students in advance that this questionnaire was about feelings and gave an example of how children might experience nervousness and anxiety, i.e., heart racing, rapid breathing, hands trembling. An example of one item from the CSAQ is as follows: “How would you feel if your teacher asked you to measure something with a ruler?” Participants are then asked to rate on an emoji scale of 1 to 5, 1 being “not nervous at all, calm”, 3 being “neither calm nor nervous”, and 5 being “very, very nervous”, how nervous they would feel about completing this task. A reliability analysis of the CSAQ conducted on the data from our sample yielded a Cronbach’s Alpha of *α* = 0.69.

#### 2.5.3. Modified Abbreviated Math Anxiety Scale (mAMAS)

The Modified Abbreviated Math Anxiety Scale (mAMAS) is based on the Abbreviated Math Anxiety Scale, an American self-report instrument developed by Hopko and colleagues (2003) [87]. It was modified by [88] to be used with British children aged 8–13 year-old and is designed to measure levels of mathematics anxiety. The scale consists of 9 items, which are rated on a 5-point Likert scale ranging from 0 (strongly disagree) to 4 (strongly agree). Respondents indicate the extent to which they experience anxiety-related thoughts and feelings when faced with mathematical tasks or situations. The mAMAS items cover various aspects of math anxiety, including worries about math classes, fear of performing poorly in math-related activities, and discomfort with mathematical problem solving. The scale aims to provide a brief yet reliable measure of math anxiety, making it suitable for use in research studies where brevity and efficiency are a priority. We used a translation of this scale previously validated for its psychometric properties by native German-speaking colleagues. Students rated their level of math anxiety on a scale of 1 to 5 with 1 being “low anxiety” and 5 being “high anxiety” based on statements such as the following: “Finding out that you are going to have a surprise math test when you start your math lesson?” Total scores on the mAMAS can range from 0 to 36, with higher scores indicating greater levels of mathematics anxiety. The reported internal consistency of this questionnaire is high. We found a Cronbach’s Alpha consistent with high reliability (*α* = 0.87) when we tested this on our sample data.

### 2.6. Procedure

Approval for the pilot study was provided by the Ethics Committee of the University of Koblenz and the state authorities in Rhineland Palatinate overseeing schools. Informed consent was sought and provided by parents and guardians of all students involved in this study. The class teacher and the principal also gave permission for this study to be conducted in the school. Students provided verbal assent prior to commencing the experiment and were informed that they could withdraw their participation with no consequences at any point. Students were tested by two female researchers in a separate classroom with access to their teacher, if required. Consistent supervision of the data collection by teaching staff could not be provided in any of the schools where data were collected and, although this arrangement was approved by the ethics committee, it may have inadvertently evoked increased anxiety in some students, an issue discussed in the limitations of this article. The room had adequate lighting and individual seating arrangements. The researchers explained the mental rotation task to the students by rotating objects such as a pair of scissors, a toy, and a wooden object, it does not change its features. Students were then asked to imagine the object in their mind then try to rotate it mentally. The purpose of this study and the significance of mental rotation in everyday life and for schoolwork was also explained. The researchers also checked in advance that students were familiar and comfortable with the use of a keyboard and a mouse. Furthermore, any student who required eye-glasses was reminded to wear these while viewing the tablet screen.

### 2.7. Data Analysis

Quantitative data analyses were conducted using SPSS^®^ 29 software. We employed a range of statistical tests to explore various aspects of our research questions. Raw scores on the math task were standardized by converting them to percentages for comparability during the statistical analyses.

The accuracy scores and response time on the mental rotation task and the spatial anxiety score variables exhibited positive skewness, as evidenced by the skewness statistic and visual inspection of the histogram and Q-Q plot. To address this, a square root transformation was applied to accuracy (Sqr_Acc) and response time (Sqr_RT) and a logarithmic transformation was applied to the spatial anxiety variable (log_CSA). This transformation improved the normality of the distribution, reducing skewness and bringing the distribution of these variables closer to a normal shape.

A Chi-Square test of association was used to test whether girls or boys were more likely to choose math or German as their preferred school subject. The assumptions for the application of this test were met, that is, the sample size was large enough, the nominal variables were categorical, participants were randomly and independently sampled, the data consisted of raw frequencies and the expected frequency (Fe) within each cell was greater than 1, and no more than 20% of the cells should have less than 5.

A multivariate analysis of covariance (MANCOVA) was used to assess gender differences in math and mental rotation performance and math and spatial anxiety. Assumptions for the MANCOVA, such as multivariate normality of the dependent variables, homogeneity of variances (verified with Levene’s test) and covariances (verified with Box’s test), linearity, homogeneity of slopes (verified by checking interaction terms), the independence of observations, and the absence of multicollinearity and outliers, were satisfied for all variables [89].

Prior to data analyses, raw skin conductance data were pre-processed by removing artefacts and noise [82]. The mean of the GSR for the three experimental conditions of baseline and during the math and mental rotation tasks was calculated for each participant in order to facilitate analysis. GSR variability can lead to skewed distributions within the data which require further processing and robust statistical methods which are not sensitive to non-normality. Following the removal of outliers and checks for normality on the skin conductance variables, the data were found to be skewed. This is common with physiological data due to the nature of measurements, which often do not follow a normal distribution [82]. Pre-processing and data transformation such as square-root and logarithmic transformation did not improve this. However, the test of normality within the MANCOVA did not reveal any significant effects. Additionally, as skin conductance can vary widely between individuals due to differences in baseline arousal levels, skin properties, and other factors, raw GSR data were standardized by performing a z-score transformation. Z-scores help normalize differences, making individual responses more comparable [90].

## 3. Results

### 3.1. Gender Differences in Math and Mental Rotation Performance, Math and Spatial Anxiety, and Subject Preference

A multivariate analysis of variance (MANOVA) was conducted to examine the effects of the fixed factor gender on the combined dependent variables of math performance (percentage scored and completion times), mental rotation performance (accuracy and response time), math anxiety, and spatial anxiety. The multivariate effect of gender was significant, *λ* = 0.80, *F*(1, 129) = 5.22, *p* < 0.001, with a small effect size of *η*^2^ = 0.20, indicating significant differences between boys and girls across the dependent variables (see Table 2).

A series of between-subjects effects tests were conducted to examine gender differences in percentage scored and completion times on the math task, accuracy and response time on the mental rotation task, math anxiety, and spatial anxiety. The analyses yielded the following results.

For the math percentage scored, there was a statistically significant effect of gender, *F*(1, 129) = 4.63, *p* = 0.033, *η*^2^ = 0.03. This indicates that gender accounts for 3% of the variance in percentage scores. Similarly, for completion times on the math task, the effect of gender was also statistically significant, *F*(1, 129) = 7.20, *p* = 0.008, *η*^2^ = 0.05, indicating that gender accounts for 5% of the variance in completion times. In contrast, gender did not have a significant effect on accuracy in the mental rotation task, *F*(1, 129) = 0.03, *p* = 0.872, *η*^2^ = 0.00, nor on response time, *F*(1, 129) = 3.20, *p* = 0.076, *η*^2^ = 0.02. The effect of gender on math anxiety was significant, *F*(1, 129) = 10.50, *p* = 0.002, *η*^2^ = 0.08. This indicates that gender explains 8% of the variance in math anxiety scores. For spatial anxiety, the effect of gender was not statistically significant, *F*(1, 129) = 2.28, *p* = 0.134, *η*^2^ = 0.02.

Overall, these results indicate significant gender differences in percentage scores and completion times on the math task, as well as in math anxiety. The effect of gender on accuracy, response time, and spatial anxiety was not significant (see Table 3).

Pairwise comparisons were conducted to examine the differences between boys and girls on various dependent variables, including math percentage scores, math completion times, accuracy, response time, math anxiety, and spatial anxiety.

For math percentage scores, girls scored significantly higher than boys, with a mean difference of 7.08, *SE* = 3.30, *p* = 0.033, 95% *CI* [0.57, 13.56]. Regarding math completion times, girls took significantly longer to complete the tasks compared to boys, *mean difference* = 2.44, *SE* = 0.91, *p* = 0.008, 95% *CI* [0.64, 4.25]. In terms of accuracy on the nMRT, there were no significant differences between boys and girls, *mean difference* = 0.005, *SE* = 0.03, *p* = 0.872, 95% *CI* [−0.05, 0.06], nor in response time, *mean difference* = −0.19, *SE* = 0.12, *p* = 0.076, 95% *CI* [−0.40, 0.02]. Regarding math anxiety, girls reported significantly higher levels of math anxiety than boys, *mean difference* = 0.43, *SE* = 0.13, *p* = 0.002, 95% *CI* [0.17, 0.69]. For spatial anxiety, there were no significant differences between boys and girls, *mean difference* = 0.005, *SE* = 0.03, *p* = 0.872, 95% *CI* [−0.05, 0.06].

Overall, the results suggest that girls outperform boys in terms of math percentage scores but take longer to complete the math task. The error bars in Figure 2 represent the standard error of the mean (SEM). The difference in the width of the error bars for percentage scored and completion times indicates that there is more variability in the scores compared to the completion times. Scores might be more sensitive to individual differences in mathematical ability, understanding, and problem-solving strategies, leading to greater variability in scores. Completion times might be less variable because they are influenced by factors such as time constraints and task pacing, which could be more consistent across participants. There are no significant differences between boys and girls in accuracy nor response time on the mental rotation task or in spatial anxiety.

A Chi-Square test for association was performed to examine the relationship between gender and preference for math vs. German. The results indicated a strong significant association between gender and subject preference, *χ*^2^ (2, *N* = 131) = 13.50, *p* < 0.001. Specifically, girls were significantly less likely than boys to indicate a preference for math and were more likely to choose German or indicate no preference for either subject (see Figure 3). A crosstabulation analysis was conducted to examine the relationship between gender (sex) and subject preference (math vs. German). The results are as follows. Out of the 66 boys, 44 (65.7%) preferred math, 8 (28.6%) preferred German, and 14 (38.9%) had no preference. Out of the 65 girls, 23 (34.3%) preferred math, 20 (71.4%) preferred German, and 22 (61.1%) had no preference. The overall distribution across both sexes showed that 67 participants (51.1%) preferred math, 28 participants (21.4%) preferred German, and 36 participants (27.5%) had no preference.

### 3.2. Association between Gender, Subject Preference, Emotional Reactivity (Measured by GSR), Subjective Anxiety (Math and Spatial Anxiety), and Performance on the Math and Mental Rotation Tasks

#### 3.2.1. Effects on Math Performance

A multivariate analysis of covariance (MANCOVA) was conducted to examine the effects of various predictors on two dependent variables: percentage scored and completion times on the math task. The predictors included two fixed factors, gender and subject preference, and three covariates, emotional reactivity measured by GSR, math anxiety, and spatial anxiety.

The results of the multivariate tests, as indicated the main effect of gender was significant, *Wilks’ Λ* = 0.91, *F*(2, 121) = 5.60, *p* = 0.004, *η*^2^ = 0.09, indicating a significant difference in the combined dependent variables based on gender. Additionally, Wilks’ Lambda showed a significant effect of math anxiety on the combined dependent variables, *Wilks’ Λ* = 0.910, *F*(2, 121) = 5.96, *p* = 0.003, *η*^2^ = 0.09, suggesting that math anxiety significantly impacts the combined dependent variables. The main effect of subject preference was also significant, *Wilks’ Λ* = 0.89, *F*(4, 242) = 3.43, *p* = 0.009, *η*^2^ = 0.05, suggesting that subject preference significantly impacts the combined dependent variables. The interaction effect between gender and subject preference was not significant, *Wilks’ Λ* = 0.98, *F*(4, 242) = 0.52, *p* = 0.718, *η*^2^ = 0.01, indicating that this interaction does not significantly affect the combined dependent variables. The individual effects of emotional reactivity measured by GSR and spatial anxiety on the combined dependent variables were not significant, with *Wilks’ Λ* = 0.97, *F*(2, 121) = 1.78, *p* = 0.172, *η*^2^ = 0.03 for emotional reactivity measured by GSR and *Wilks’ Λ* = 0.97, *F*(2, 121) = 1.93, *p* = 0.15, *η*^2^ = 0.03 for spatial anxiety.

Overall, these multivariate tests demonstrate that gender has a significant impact on the dependent variables, with notable contributions from subject preference and math anxiety. The effect of emotional reactivity measured by GSR and spatial anxiety were not significant. The results are summarized in Table 4.

A series of tests of between-subjects effects were conducted to examine the influence of gender, subject preference on percentage scored, and completion times on the math task while controlling for emotional reactivity measured by GSR and math anxiety and spatial anxiety. The analyses yielded the following results.

For the dependent variable percentage scored, the between-subjects effects indicated several significant results. There was a significant effect of math anxiety on percentage scored, *F*(1, 122) = 11.86, *p* < 0.001, *η*^2^ = 0.09. The effect of gender on percentage scored was also significant, *F*(1, 122) = 6.53, *p* = 0.012, *η*^2^ = 0.051.

For the dependent variable completion times, the between-subjects effects showed the following significant results. The effect of gender on completion times was significant, *F*(1, 122) = 4.44, *p* = 0.037, *η*^2^ = 0.03. There was also a significant effect of subject preference on completion times, *F*(2, 122) = 5.94, *p* = 0.003, *η*^2^ = 0.09. The effect of emotional reactivity measured by GSR on completion times was not significant, *F*(1, 122) = 3.06, *p* = 0.083, *η*^2^ = 0.02, nor was the effect of spatial anxiety, *F*(1, 122) = 3.79, *p* = 0.054, *η*^2^ = 0.03. The interaction effect between gender and subject preference was not significant for either percentage scored, *F*(2, 122) = 1.01, *p* = 0.367, *η*^2^ = 0.02, or completion times, *F*(2, 122) = 0.08, *p* = 0.921, *η*^2^ = 0.00.

These results indicate that, while gender and math anxiety have significant effects on math percentage scored, gender and subject preference are significant for completion times. The interaction between gender and subject preference does not significantly impact either dependent variable. Additionally, emotional reactivity measured by GSR and spatial anxiety do not show significant effects on math completion times (see Table 5).

#### 3.2.2. Gender-Specific Effects on Math Performance

A multivariate analysis of covariance (MANCOVA) was conducted to examine the effects of subject preference, emotional reactivity measured by GSR, math anxiety, and spatial anxiety on math percentage scored and completion times separately for boys and girls.

For boys:

Multivariate tests showed that the effects of emotional reactivity measured by GSR on boys’ performance on the math task were not significant (*Wilks’ Λ* = 0.10, *F*(2, 59) = 0.09, *p* = 0.915, *η*^2^ = 0.00), neither was math anxiety (*Wilks’ Λ* = 0.95, *F*(2, 59) = 1.55, *p* = 0.220, *η*^2^ = 0.05), nor spatial anxiety (*Wilks’ Λ* = 0.98, *F*(2, 59) = 0.72, *p* = 0.489, *η*^2^ = 0.02), nor subject preference (*Wilks’ Λ* = 0.90, *F*(4, 118) = 1.62, *p* = 0.173, *η*^2^ = 0.05). Tests of between-subjects effects showed no significant effect of emotional reactivity on boys’ percentage scored (*F*(1, 60) = 0.01, p = 0.929, *η*^2^ = 0.00). Neither math anxiety (*F*(1, 60) = 3.14, *p* = 0.081 *η*^2^ = 0.05), nor spatial anxiety (*F*(1, 60) = 0.02, *p* = 0.876, *η*^2^ = 0.00), nor subject preference (*F*(2, 60) = 1.04, *p* = 0.359, *η*^2^ = 0.03) showed significant effects on percentage scored on the math task.

For boys’ completion times on the math task, neither emotional reactivity measured by GSR (*F*(1, 60) = 0.172, *p* = 0.680, *η*^2^ = 0.00), nor math anxiety (*F*(1, 60) = 0.02, *p* = 0.878, *η*^2^ = 0.00), nor spatial anxiety (*F*(1, 60) = 1.44, *p* = 0.234, *η*^2^ = 0.02), nor subject preference (*F*(2, 60) = 2.31, *p* = 0.108, *η*^2^ = 0.07) showed significant effects.

For girls:

Multivariate tests showed significant effects of emotional reactivity measured by GSR (*Wilks’ Λ* = 0.84, *F*(2, 58) = 5.53, *p* = 0.006, *η*^2^ = 0.16) on girls’ performance on the math task. Math anxiety (*Wilks’ Λ* = 0.83, *F*(2, 58) = 5.93, *p* = 0.005, *η*^2^ = 0.17) and subject preference (*Wilks’ Λ* = 80, *F*(4, 116) = 3.45, *p* = 0.011, *η*^2^ = 0.11) also had significant effects, but spatial anxiety did not (*Wilks’ Λ* = 0.95, *F*(2, 58) = 1.67, *p* = 0.197, *η*^2^ = 0.05) (see Table 6).

Tests of between-subjects effects showed that emotional reactivity measured by GSR had no significant effects on girls’ percentage scored, *F*(1, 59) = 1.34, *p* = 0.251, *η*^2^ = 0.02. Math anxiety showed significant effects (*F*(1, 59) = 10.22, *p* = 0.002, *η*^2^ = 0.145), but spatial anxiety (*F*(1, 59) = 1.13, *p* = 0.292, *η*^2^ = 0.02) and subject preference (*F*(2, 59) = 1.45, *p* = 0.242, *η*^2^ = 0.05) did not (see Table 7).

For girls’ completion times, emotional reactivity measured by GSR showed significant effects (*F*(1, 59) = 8.48, *p* = 0.005, *η*^2^ = 0.13), as did subject preference (*F*(2, 59) = 5.43, *p* = 0.007, *η*^2^ = 0.16), but math anxiety (*F*(1, 59) = 0.65, *p* = 0.423, *η*^2^ = 0.01) and spatial anxiety (*F*(1, 59) = 2.76, *p* = 0.102, *η*^2^ = 0.04) did not (see Figure 4).

For boys, none of the predictors were significant for performance on the math task. For girls, math anxiety had a significant effect on percentage scored, and both emotional reactivity and subject preference had significant effects on completion times. For girls, lower math anxiety is significantly associated with higher math scores, and reduced emotional reactivity is significantly associated with shorter completion times on the math task. Subject preference does not significantly affect their math scores but girls who preferred math also had significantly shorter completion times.

#### 3.2.3. Effects on Mental Rotation Performance

A multivariate analysis of covariance (MANCOVA) was conducted to examine the associations between the fixed factors gender, subject preference, and performance on the combined dependent variables accuracy and response time on the mental rotation task when controlling for covariates of emotional reactivity measured by GSR and math and spatial anxiety. The results of the multivariate tests, as indicated by Wilks’ Lambda, showed no significant effect of emotional reactivity measured by GSR on mental rotation performance, *Wilks’ Λ* = 0.96, *F*(2, 121) = 2.70, *p* = 0.071, *η*^2^ = 0.04. Neither math anxiety, *Wilks’ Λ* = 0.96, *F*(2, 121) = 2.64, *p* = 0.075, *η*^2^ = 0.04, nor spatial anxiety, *Wilks’ Λ* = 0.10, *F*(2, 121) = 0.23, *p* = 0.798, *η*^2^ = 0.004, had a significant effect. A significant effect of gender, *Wilks’ Λ* = 0.94, *F*(2, 121) = 3.73, *p* = 0.027, *η*^2^ = 0.06, but no significant effect of subject preference, *Wilks’ Λ* = 0.96, *F*(4, 242) = 1.11, *p* = 0.350, *η*^2^ = 0.02, was found. There was no significant interaction effect for gender and subject preference, *Wilks’ Λ* = 0.98, *F*(4, 242) = 0.71, *p* = 0.582, *η*^2^ = 0.01 (see Table 8).

The results of multivariate tests further indicate that gender had a significant effect on accuracy and response time in the mental rotation task, while emotional reactivity measured by GSR and math anxiety both showed no significant effects.

A series of between-subjects effect tests were conducted to examine the influence of gender, emotional reactivity measured by GSR, math anxiety, and spatial anxiety on accuracy and response time on the mental rotation task. The analyses yielded the following results.

For accuracy, the between-subjects effects indicated the following results. The effect of emotional reactivity measured by GSR was not significant, *F*(1, 122) = 0.27, *p* = 0.603, *η*^2^ = 0.00, but the effect of math anxiety was significant, *F*(1, 122) = 5.26, *p* = 0.023, *η*^2^ = 0.04. The effects of spatial anxiety, *F*(1, 122) = 0.310, *p* = 0.579, *η*^2^ = 0.003, gender, *F*(1, 122) = 0.02, *p* = 0.885, *η*^2^ = 0.00, and subject preference, *F*(2, 122) = 0.138, *p* = 0.871, *η*^2^ = 0.00, on accuracy were not significant.

For response time, the between-subjects effects showed the following results. The effect of emotional reactivity measured by GSR was significant, *F*(1, 122) = 5.43, *p* = 0.021, *η*^2^ = 0.04. The effects of math anxiety, *F*(1, 122) = 0.80, *p* = 0.373, *η*^2^ = 0.01, and spatial anxiety, *F*(1, 122) = 0.274, *p* = 0.602, *η*^2^ = 0.00, were not significant. The effect of gender was significant, *F*(1, 122) = 7.12, *p* = 0.009, *η*^2^ = 0.06, but subject preference was not, *F*(2, 122) = 2.06, *p* = 0.131, *η*^2^ = 0.03 (see Table 9).

These results indicate that math anxiety has a significant effect on accuracy, while emotional reactivity measured by GSR and gender have significant effects on response time. However, neither spatial anxiety, nor subject preference, nor their interaction have significant effects on mental rotation performance.

#### 3.2.4. Gender-Specific Effects on Mental Rotation Performance

The results of a MANCOVA examining the gender-specific effects of emotional reactivity, math anxiety, and spatial anxiety, as well as subject preference on accuracy and response time on the mental rotation task, are presented below.

For boys:

Multivariate tests showed no significant effect of emotional reactivity measured by GSR on boys’ mental rotation performance (*Wilks’ Λ* = 0.95, *F*(2, 59) = 1.57, *p* = 0.217, *η*^2^ = 0.050). Neither math anxiety (*Wilks’ Λ* = 0.94, *F*(2, 59) = 1.74, *p* = 0.185, *η*^2^ = 0.06), spatial anxiety (*Wilks’ Λ* = 0.99, *F*(2, 59) = 0.36, *p* = 0.699, *η*^2^ = 0.01), nor subject preference (*Wilks’ Λ* = 0.912, *F*(4, 118) = 1.40, *p* = 0.239, *η*^2^ = 0.04) show significant effects.

Tests of between-subjects effects showed no significant effects of emotional reactivity on boys’ accuracy (*F*(1, 60) = 0.61, *p* = 0.437, *η*^2^ = 0.01). Neither math anxiety (*F*(1, 60) = 3.35, *p* = 0.072, *η*^2^ = 0.05), nor spatial anxiety (*F*(1, 60) = 0.003, *p* = 0.954, *η*^2^ = 0.00), nor subject preference (*F*(2, 60) = 0.31, *p* = 0.734, *η*^2^ = 0.01) showed significant effects.

For boys’ response time, neither emotional reactivity measured by GSR (*F*(1, 60) = 3.17, *p* = 0.080, *η*^2^ = 0.05), nor math anxiety (*F*(1, 60) = 0.08, *p* = 0.783, *η*^2^ = 0.00), nor spatial anxiety, (*F*(1, 60) = 0.60, *p* = 0.442, *η*^2^ = 0.01) nor subject preference (*F*(2, 60) = 1.91, *p* = 0.156, *η*^2^ = 0.06) showed significant effects.

For girls:

Multivariate tests showed no significant effect of emotional reactivity measured by GSR on girls’ mental rotation performance *(Wilks’ Λ* = 0.94, *F*(2, 58) = 1.70, *p* = 0.191, *η*^2^ = 0.05). Neither math anxiety (*Wilks’ Λ* = 0.97, *F*(2, 58) = 0.974, *p* = 0.384, *η*^2^ = 0.03), nor spatial anxiety (*Wilks’ Λ* = 0.98, *F*(2, 58) = 0.72, *p* = 0.490, *η*^2^ = 0.02), nor subject preference (*Wilks’ Λ* = 0.98, *F*(4, 116) = 0.28, *p* = 0.893, *η*^2^ = 0.01) showed significant effects.

Tests of between-subjects effects showed no significant effects of emotional reactivity on girls’ accuracy, *F*(1, 59) = 0.34, *p* = 0.562, *η*^2^ = 0.01), and neither did math anxiety (*F*(1, 59) = 1.37, *p* = 0.246, *η*^2^ = 0.02), nor spatial anxiety (*F*(1, 59) = 1.28, *p* = 0.262, *η*^2^ = 0.02), nor subject preference (*F*(2, 59) = 0.12, *p* = 0.891, *η*^2^ = 0.00).

For girls’ response times, neither emotional reactivity measured by GSR (*F*(1, 59) = 2.57, *p* = 0.114, *η*^2^ = 0.04), nor math anxiety (*F*(1, 59) = 1.02, *p* = 0.316, *η*^2^ = 0.02), nor spatial anxiety (*F*(1, 59) = 0.03, *p* = 0.854, *η*^2^ = 0.00), nor subject preference (*F*(2, 59) = 0.478, *p* = 0.623, *η*^2^ = 0.00) showed significant effects.

In summary, none of the factors showed significant effects on spatial performance for either gender.

## 4. Discussion

### 4.1. Overview of Findings

This study explored gender differences in mathematics and mental rotation performance, focusing on the roles of subjective anxiety (math and spatial anxiety), subject preference math versus German, and emotional reactivity as measured by the Galvanic Skin Response (GSR). The findings indicate that girls outperformed boys in percentage scores on the math task but took longer to complete it. This aligns with previous research suggesting that girls often perform just as well or better than boys in terms of scores but take more time, possibly due to higher levels of anxiety [35,40]. Lower math anxiety in girls was associated with higher percentage scores, highlighting the critical role of anxiety in cognitive performance [43,91]. However, no significant gender differences were found in mental rotation accuracy nor response times, suggesting that spatial ability might be more evenly distributed among primary school children when tasks are designed to mitigate stereotype threat [22,23,24,75].

### 4.2. Gender Differences in Math and Mental Rotation Performance, Subjective Anxiety, and Subject Preference

The MANOVA results demonstrated significant gender differences in math performance, with girls scoring higher but taking longer to complete the tasks. This may be due to the cognitive load imposed by higher anxiety levels, which can limit working memory capacity and slow down processing speed [34,35]. Girls’ higher math anxiety, consistent with previous findings [77,91], likely contributed to their longer completion times. This anxiety may cause girls to adopt a more meticulous approach to problem solving, resulting in higher accuracy but slower performance [36,38]. Girls may have taken extra time to thoroughly understand and accurately solve each problem, leading to fewer mistakes and higher overall scores. This strategy contrasts with potentially quicker, but less precise, approaches and has been found to produce better results in children [38]. Boys were faster but scored lower on the math task, which might be explained by a tendency to prioritize speed over accuracy. This approach may reflect social validation sought by boys from peers regarding their math ability [36]. This involves rushing through problems to complete the task quickly, which can lead to more mistakes and lower overall scores. Additionally, boys might have been more confident in their initial responses and less likely to double-check their work, resulting in a higher rate of errors. This contrasts with a more thorough approach, where taking additional time to carefully solve each problem can lead to higher accuracy and better performance, a strategy also found to be effective on spatial tasks [37].

Interestingly, no significant gender differences were found in spatial anxiety, contrasting with some studies that report higher spatial anxiety in girls [41,55]. The superior math performance of girls in this study may have mitigated the expected differences in spatial anxiety, suggesting that mathematical self-efficacy might mitigate anxiety levels across related cognitive domains [13]. Furthermore, some studies show that girls excel in the Kangaroo Challenge, in which the math problems involved are known to correlate with spatial abilities [92,93]. Additionally, the order of the math task, which preceded the spatial task, demonstrates that practice on number line estimation, word problems, and missing term tasks, known to correlate with mental rotation, may also have reduced students’ spatial anxiety [94,95]. From a statistical perspective, the spatial anxiety variable in the current study, although improved through log transformation, deviated marginally from normality. The potential effects of this are discussed further in the limitations.

Contrary to traditional findings that boys outperform girls on mental rotation tasks [96], this study found no significant gender differences in accuracy or response times on the nMRT. This result supports recent research suggesting that, when tasks are designed to minimize stereotype threat, gender differences in spatial abilities may diminish [20,23]. The use of gender-congruent and -neutral stimuli and the inclusion of a balanced task design likely contributed to this outcome [22,24]. As observed by the researchers, a preference expressed by children for the tablet-based task over the paper-and-pencil math task also highlights the potential impact of task format on engagement and performance.

The significant association between gender and subject preference, with girls less likely to choose math as their preferred subject, reflects enduring societal stereotypes and their influence on academic choices [12,13,14]. This preference for German over math among girls may stem from their higher math anxiety and the internalization of societal beliefs about gender and mathematical ability [9,10].

### 4.3. Association between Gender, Subject Preference, Emotional Reactivity Measured by GSR, Subjective Anxiety, and Performance on the Math and Mental Rotation Tasks

A unique aspect of the present study is that it explored the association between gender and subject preference, but also multiple related psychological as well as physiological factors and their effects on performance on math and mental rotation tasks among primary school children. The findings provide insights into the role of these factors in task performance, with notable differences observed across gender and task type.

#### 4.3.1. Subject Preference

Subject preference significantly impacted math completion times but not scores. Students who preferred math completed tasks quicker, highlighting the importance of interest and engagement in efficient performance. This supports the hypothesis and aligns with research suggesting that subject interest, societal stereotypes, and intrinsic motivation significantly influence academic performance [12,14,97]. Additionally, this result provides further evidence for the reciprocal effects of engagement and interest on math self-concept, which in turn leads to greater achievement, and vice versa [98,99].

The finding that a preference for math over German had a protective effect on completion times for the math task, particularly among girls, is noteworthy. This suggests that girls who prefer math may experience reduced cognitive and emotional barriers when engaging with mathematical tasks. One possible explanation is that these girls may be less susceptible to stereotype threat. By identifying more strongly with math, these girls may be less affected by societal stereotypes that suggest girls are less capable in math, thereby mitigating the anxiety that can deplete working memory and hinder task performance [34]. Girls with a preference for math might view the challenge of the task as an opportunity to improve rather than as threats to their self-concept, which could further reduce performance anxiety and increase task efficiency. Additionally, a genuine interest and enjoyment in math could create a positive feedback loop where pleasure and fun derived from engaging in the subject counteract the stress typically associated with challenging tasks [97,100]. When girls enjoy math, they are likely to approach the task with greater confidence and motivation, which can enhance focus and efficiency, leading to faster completion times. Another potential factor could be the development of more effective problem-solving strategies. Girls who prefer math might have more practice and experience with mathematical thinking, allowing them to navigate the tasks more quickly eliciting less cognitive load [43]. These factors work together to create a more favorable psychological environment for girls, enabling them to perform math tasks more efficiently.

#### 4.3.2. Emotional Reactivity

The results indicate that emotional reactivity, as measured by GSR, plays a nuanced role in mathematical and spatial task performance. For the math task, higher emotional reactivity was a significant predictor of longer completion times but did not significantly impact the percentage scored. This suggests that, while physiological arousal may slow down task completion, it does not necessarily detract from the precision of performance. Similarly, emotional reactivity was a significant predictor of response time on the mental rotation task, indicating that higher physiological arousal was associated with slower performance. These findings align with the Yerkes–Dodson law, which posits that higher levels of arousal can hamper performance on complex tasks by decreasing focus and cognitive resources [60,101]. This also provides evidence to support the theory that executive resources such as working memory are limited by emotional reactivity, leading to interference with cognitive focus, slowing down responses on both tasks [34,95].

A particular significance of this study, therefore, lies in its contribution to understanding the physiological underpinnings of how negative emotions and stereotype thinking impact cognitive performance. This is particularly relevant for females, who are often subjected to stereotypes about their abilities in mathematics and spatial tasks and related fields [102]. The findings highlight that the physiological arousal associated with stereotype threat not only affects how quickly tasks are completed but also suggests a potential mechanism by which stereotype threat undermines cognitive performance. When females face tasks in domains where they are stereotypically expected to perform poorly, the resulting emotional arousal may deplete the cognitive resources needed for optimal performance, thereby reinforcing the stereotype [34,103].

This study adds important empirical evidence to the literature on gender differences in cognitive performance by demonstrating that emotional reactivity, an often-overlooked factor, plays a significant role in these differences. By measuring physiological arousal through GSR, this study provides concrete data showing that the emotional and physiological responses elicited by stereotype threat can slow performance, even if accuracy remains unaffected. This finding is critical for educators and policymakers aiming to address gender disparities in STEM fields, as it suggests that interventions designed to reduce stereotype threat and manage emotional reactivity could be key to improving performance outcomes for girls in mathematics and spatial tasks.

Additionally, the findings underscore the complex interplay between emotional states and cognitive performance, particularly under conditions of stereotype threat. This study not only broadens the understanding of how gender differences in performance manifest, but also points to the importance of considering physiological and emotional factors in educational interventions. By emphasizing the role of emotional reactivity, this research makes a vital contribution to knowledge on how gender disparities in STEM fields can be mitigated.

#### 4.3.3. Subjective Anxiety

Math anxiety emerged as a significant predictor of performance on both the math and the mental rotation tasks, with higher levels of anxiety associated with lower math percentage scores and less mental rotation accuracy. The former is consistent with the existing literature highlighting the detrimental effects of math anxiety on math performance, likely due to increased cognitive load and interference with working memory [34,91,95]. Similar to research on the effects of psychological factors such as math self-concept on accuracy on a mental rotation task [13], the findings of the current study suggest that individuals with higher anxiety about math also tend to struggle more with spatial tasks like mental rotation.

Interestingly, in this study, math anxiety did not significantly affect math completion times nor mental rotation response time. This suggests that the primary impact of math anxiety may be on the precision and accuracy of math and spatial task performance rather than on the speed of processing. This distinction is important because it indicates that, while anxious individuals may not necessarily take longer to complete tasks, their heightened anxiety could lead to more errors or less accurate responses. This could be due to the fact that anxiety diverts cognitive resources away from the task at hand, leading to a focus on avoiding mistakes rather than on efficient problem solving [37,39].

In contrast, spatial anxiety did not emerge as a significant predictor of either percentage scores or completion times on the math task nor accuracy and response times on the spatial task. This finding is intriguing, particularly in light of the literature discussed in the introduction, which suggests that spatial anxiety can disrupt performance on tasks that require spatial reasoning, such as mental rotation [41,55,56]. One possible explanation for this result is that the math problems, although correlating with mental rotation, were mathematical in nature, thus diminishing the influence of spatial anxiety on performance outcomes. Additionally, the prior engagement with these kinds of math problems may have served as a warm-up that reduced the impact of spatial anxiety on subsequent spatial performance. The lack of a significant relationship between spatial anxiety and performance in this study could also reflect the influence of other moderating factors, such as task familiarity or the presence of stereotype threat, which were mentioned in the introduction as important considerations in understanding the effects of anxiety on cognitive performance.

Overall, these findings contribute to a nuanced understanding of the role of anxiety in educational contexts, highlighting that, while math anxiety clearly impairs performance accuracy, its effects on speed are less straightforward. This insight underscores the importance of considering the specific cognitive demands of tasks when assessing the effects of different types of anxiety on performance, as well as the potential for targeted interventions to mitigate these effects.

### 4.4. Gender-Specific Effects of Subject Preference, Emotional Reactivity, and Subjective Anxiety on Math and Mental Rotation Performance

This study revealed significant gender differences in how subject preferences, emotional reactivity, and subjective anxiety impact task performance. For boys, none of the predictors played a significant role in performance outcomes on the math tasks. These findings may indicate that boys’ performance is less affected by anxiety and physiological arousal, or it may reflect different coping mechanisms or cognitive strategies employed by boys [54,73]. Additionally, boys may also be aware of stereotypes suggesting male superiority in math, which could increase pressure to excel in math tasks. This pressure to uphold the stereotype and compete with male peers may lead to faster performance but result in more errors [37]. Additionally, an element of bravado along with peer pressure may lead boys to under-report subjective anxiety [36,104]. On the mental rotation task, none of the factors were significantly associated with boys’ spatial performance.

Girls, on the other hand, performed better on the math task when they reported less subjective anxiety, that is, they had better percentage scores. Additionally, their preference for math, and an ability to keep physiological arousal in check and regulate emotions, was associated with shorter completion times on the same task. None of the factors played a role in girls’ performance on the mental rotation task. These findings suggest that girls’ math performance is more sensitive to the effects of subject preference, anxiety, and physiological arousal. There are several possible explanations for this difference, which have been previously identified in the literature. Firstly, females are known to employ an analytic, piecemeal processing strategy when attempting to solve items on a task with spatial elements [102], such as the Kangaroo problems. This approach is known to be more time-consuming than the alternative holistic approach employed by males. It is well-known that having sufficient time is an important factor for females in performing spatial tasks [105]. Also, higher sensitivity to emotional and physiological states could be attributed to gender differences in emotion regulation strategies and stress responses [73,74]. Stereotypes associated with female math inferiority, in other words, stereotype threat, may have had a more powerful effect for girls’ performance on the math task than on the less familiar mental rotation task [9,51,54]. Moreover, evidence from research has found that individuals with high levels of math anxiety and increased physiological arousal showed decreases in math accuracy compared to those who had lower math-related anxiety [48].

Girls who indicated a preference for math had significantly higher percentage scores and lower completion times on the math task. This preference for math reflects a characteristic of the sample and provides further evidence that reciprocal effects exist between engagement, practice, a more positive math attitude, and performance on both mathematical tasks [13,99,106].

As mentioned, neither boys’ nor girls’ mental rotation performance was significantly influenced by any of the factors measured in this study, including emotional reactivity, math anxiety, or spatial anxiety. This finding is noteworthy, particularly in light of the well-documented gender differences in mental rotation tasks, where boys have traditionally outperformed girls [96]. However, recent research suggests that, when the design of such tasks is adjusted to reduce gender bias and stereotype threat, these differences may diminish or even disappear [20,22,23,24]. By carefully selecting stimuli that were gender-congruent or gender-neutral and ensuring that the task was novel and free from culturally ingrained biases, this study likely reduced the activation of any stereotype-related concerns. This might explain why none of the measured factors, including those typically associated with stereotype vulnerability, such as math and spatial anxiety, had a significant impact on performance [34,41,51]. Additionally, the extended time limits may have allowed for more thoughtful and strategic processing, further reducing the influence of anxiety or emotional reactivity. The opportunity to approach the task without the pressure of a strict time constraint might have enabled both boys and girls to employ their optimal cognitive strategies, leading to similar levels of performance across genders. This outcome supports the idea that time constraints and task pressure are critical factors that can exacerbate or mitigate gender differences in spatial tasks [35]. This highlights the importance of task design in educational assessments and suggests that gender differences in spatial abilities might be more context-dependent than previously assumed. Overall, the lack of significant effects on children’s spatial performance in this study provides valuable insights into how stereotype threat can be mitigated through thoughtful task design, and it challenges the notion that gender differences in spatial abilities are innate or unchangeable.

### 4.5. Limitations and Future Research

Generally, there is a need to replicate this study in future research due to a number of limitations outlined in this section. The log-transformed spatial anxiety variable was approximately normally distributed, with minor deviations that did not significantly impact the analyses’ validity. However, caution is advised regarding these deviations as they can lead to the incorrect interpretation of related findings.

Physiological measurements, such as skin conductance, inherently vary and can be skewed. Despite standardization and normality tests, some limitations persist due to individual differences in baseline levels and potential measurement artifacts [82]. Thus, while our findings offer valuable insights, further research is needed to confirm these results and explore their generalizability across different populations. Other physiological measurements such as pupillometry, eye-tracking measurement, or photoplethysmography (PPG) could capture autonomic nervous system activity during cognitive tasks, which may provide more insight into the effects of physiological arousal.

The math task used in this study, derived from the Kangaroo Challenge, differs from the standardized problems children typically encounter in the German primary school curriculum. Although designed to engage and encourage enjoyment of math [83], the unfamiliar and creative nature of these problems might have caused stress or led to rushing through the task to engage with the computerized spatial task.

Future studies could benefit from utilizing a computerized math task. Such a task, which also records the response times of individual items, would allow for a full investigation of the effects of gender, subject preferences, subjective anxiety, and physiological arousal on performance on more difficult tasks, for example. This could facilitate an analysis of whether gender differences in scores and processing times persist on more difficult math tasks. An analysis of the current paper-and-pencil data allowed for only a limited insight into the effects of physiological arousal and the math completion times as a whole rather than individual items.

A notable limitation is the reliance on quantitative data, which does not capture the nuanced experiences and thoughts of the children during the tasks. Including qualitative data could provide deeper insights into the cognitive and emotional processes children experience while performing math and mental rotation tasks. Mixed methods allow for the integration of both quantitative (physiological measures, test scores) and qualitative (e.g., interviews, observations) data, providing a more comprehensive understanding of the phenomena under study. This approach helps to capture the complexity of children’s experiences with anxiety and cognitive tasks, offering insights that might be missed with a single-method approach.

The sample may be biased because the children who agreed to participate likely had more interest and enjoyment in math than the general school population. Furthermore, their parents may have consented for children with a more positive attitude towards math after discussing participation in this study with them. Additionally, the sample was taken from one region in a single European WEIRD (Western, Educated, Industrialized, Rich, and Democratic) state. Replicating this study with a more diverse socio-economic and cultural population may contribute to a higher generalizability of the findings to primary school contexts globally.

A factor which may have led to increased anxiety levels in students is testing by unfamiliar individuals which did not take place in their own classroom. Although they had access to their teachers at any stage, this may have caused some children to feel more anxious. Moreover, the effects of teacher and educator attitudes and anxieties regarding math and spatial activities were not explored in this study, but could have valuable educational implications. Information about parent’s socio-economic status and cultural background could also enhance findings in future research.

Future research should investigate the associations between anxiety and physiological arousal with larger, more diverse samples to confirm and elaborate on these findings. Longitudinal studies could offer insights into how these relationships evolve over time and across different educational stages. Additionally, examining the effectiveness of specific interventions such as mindfulness or relaxation in primary education aimed at reducing anxiety and managing physiological arousal could offer practical solutions for improving academic performance.

### 4.6. Educational Implications of This Study

The results of this study suggest several implications for teachers and educators. Given that girls demonstrate higher math performance but also higher anxiety in mathematics, interventions should focus on reducing math anxiety and providing support to build confidence. Encouraging a positive math identity in girls from a young age could help balance subject preferences and potentially increase the number of girls pursuing STEM fields in the future [97]. Furthermore, fostering interest and enthusiasm for math in girls can have significant benefits for their performance [100]. This could involve providing positive role models, offering engaging math activities, and addressing anxiety through targeted interventions [107]. Engaging and relevant math problems that connect to real-world applications can make the subject more relatable and interesting [92,93]. Interventions such as stress management training, mindfulness practices, and anxiety reduction techniques can help students manage physiological arousal and negative emotions, thereby enhancing performance. For boys, strategies to improve focus and attention during tasks may be more effective. Also, techniques such as positive reinforcement and encouraging a growth mindset can also be beneficial [51,108]. Educators could counteract gender stereotypes in math and science by promoting positive role models and creating an inclusive classroom environment. Encouraging all students to see themselves as capable mathematicians can improve self-concept and reduce anxiety [8,100]. Overall, these educational strategies could contribute to more balanced academic outcomes and encourage greater participation of girls in STEM disciplines.

Additionally, as spatial ability is critical in many STEM-related tasks, such as geometry, engineering design, architecture, and physics, the spill-over effects of math-related psychological and emotional factors such as math self-concept, math anxiety, and physiological arousal negatively impact spatial abilities. Students with high math anxiety and high emotional reactivity may struggle in these areas, even if their spatial reasoning skills would otherwise be strong. This could result in students avoiding STEM subjects or careers, perpetuating under-representation in fields that rely heavily on spatial reasoning. Curriculums could integrate more spatial reasoning activities in non-math contexts, which would allow students to develop spatial skills without the pressure of math tasks, potentially reducing the impact of math anxiety. For example, including more visual arts, design, and hands-on learning can improve spatial reasoning in less anxiety-inducing environments [109].

## 5. Conclusions

This study highlights significant gender differences in mathematics performance, subjective anxiety, subject preference, and emotional reactivity among primary school children. Girls outperformed boys in percentage scores on the math task but took longer to complete it, suggesting a more meticulous approach with higher anxiety levels playing a significant role. Boys, on the other hand, prioritized speed over precision, resulting in lower scores but faster completion times. Math anxiety significantly impacted students’ performance, with higher anxiety associated with lower scores for both genders, though its effect was more pronounced in girls. Emotional reactivity, as measured by GSR, influenced completion times for the math and the spatial tasks, indicating that higher physiological arousal can slow performance on both. Girls who preferred math performed better in this domain and completed the task faster, highlighting the importance of fostering a positive attitude towards the subject at this important educational stage in order to enhance performance. No significant gender differences were found in mental rotation accuracy nor response times, suggesting that spatial ability may be more evenly distributed among primary school children or that educational practices in Germany effectively mitigate these differences. These findings underscore the need for educational interventions designed to address the unique needs of boys and girls.

Furthermore, addressing societal influences and systemic issues such as gender stereotypes, poor attitudes towards mathematics, and gender inequity in STEM fields is vital for creating a more equitable and supportive environment for all students. Several interventions can be implemented; for example, training teachers in gender-sensitive practices and diverse teaching methods [110]; developing educational materials that promote gender equality and feature diverse STEM role models [111]; engaging parents and communities through workshops and resources to support positive attitudes towards math and STEM [112]; launching media campaigns to change societal perceptions and highlight successful women in STEM [113]; advocating for policies that promote gender equity in education, including training for educators and anti-discrimination policies [114]; and conducting research to continuously assess and improve these interventions. These steps aim to create a more inclusive and supportive environment, benefiting all students and fostering a culture of respect and equity in education.

## Figures and Tables

**Figure 1 behavsci-14-00809-f001:**
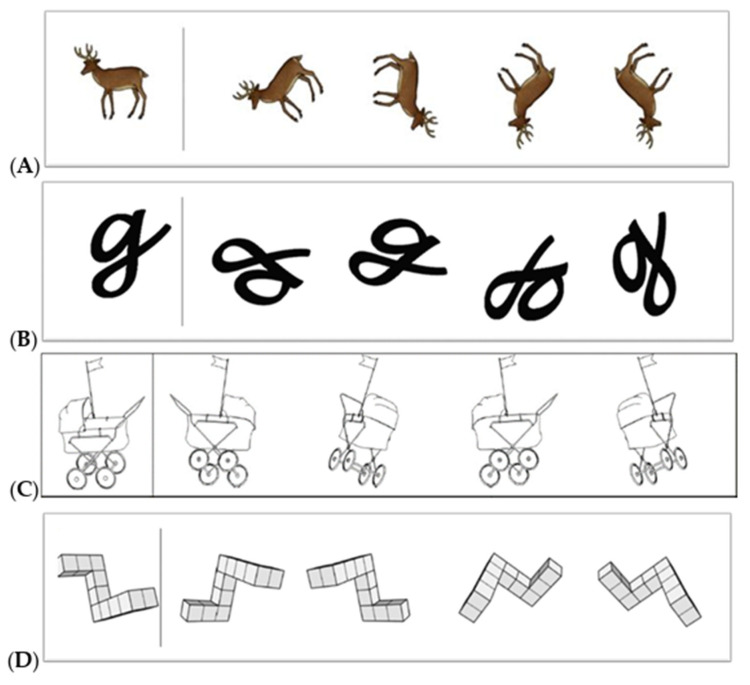
Examples of MRT items used with concrete objects: (**A**) animal stimulus and (**B**) letter stimulus [85]; (**C**) female-stereotyped stimulus “Pram” [21,23]; and with an abstract object (**D**) “Cube” figure [22].

**Figure 2 behavsci-14-00809-f002:**
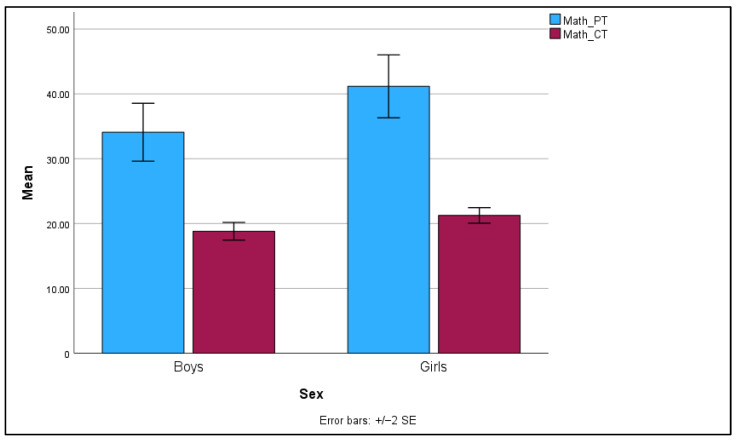
Gender (sex) differences in percentage scored (blue bar = Math_PT) and completion times (red bar = Math_CT) on the math task. Error bars represent the standard error of the mean (SEM). The error bars indicate similar variability in scores and completion times between boys and girls.

**Figure 3 behavsci-14-00809-f003:**
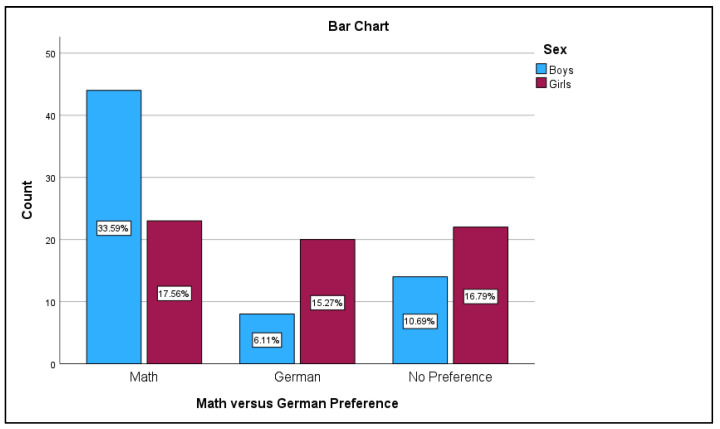
Percentage of math vs. German as the preferred school subject by gender (sex).

**Figure 4 behavsci-14-00809-f004:**
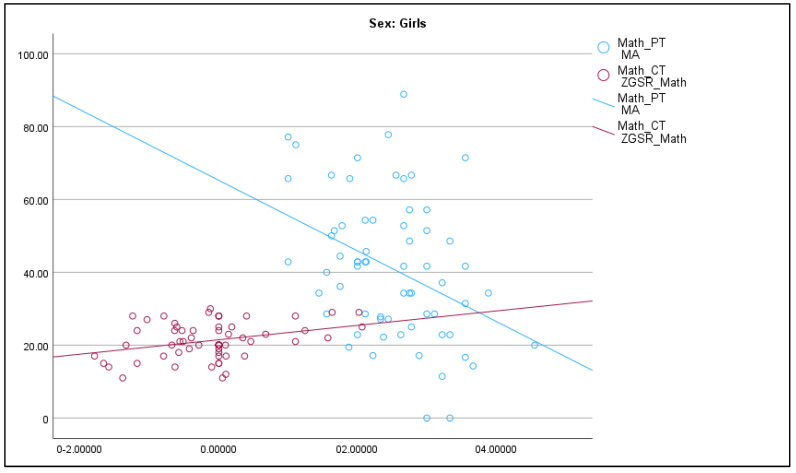
Effects of math anxiety on percentage scores and emotional reactivity (GSR) on completion times for girls. X-Axis = standardized scores; Y-Axis = percentage scores and completion times; blue circles: percentage scores (Math_PT) vs. math anxiety (MA); blue line: trend line for percentage scores (Math_PT); red circles: completion times (Math_CT) vs. emotional reactivity (ZGSR_Math); red line: trend line for completion times (Math_CT).

**Table 1 behavsci-14-00809-t001:** Class, sex crosstabulation, and grade in primary school.

		Sex		Total
		Boys	Girls	
Class	2nd Grade	16	21	37
	3rd Grade	23	22	45
	4th Grade	27	22	49
Total		66	65	131

**Table 2 behavsci-14-00809-t002:** Multivariate tests: effect of gender (sex) on performance on math and mental rotation tasks, math, and spatial anxiety in primary school children.

Effect		Value	F	Hypothesis df	Error df	Sig.	Partial Eta Squared
Sex	Wilks’ Lambda	0.798	5.219	6.000	124.000	<0.001	0.202

Design: intercept + sex.

**Table 3 behavsci-14-00809-t003:** Tests of between-subjects: effects of gender (sex) on percentage scored (Math_PT) and completion time (Math_CT) on the math task, and math anxiety (MA) in primary school children.

Source	Dependent Variable	Type III Sum of Squares	df	Mean Square	F	Sig.	Partial Eta Squared
Sex	Math_PT	1643.461	1	1643.461	4.630	0.033	0.035
	Math_CT	195.693	1	195.693	7.200	0.008	0.053
	MA	6.015	1	6.015	10.501	0.002	0.075

**Table 4 behavsci-14-00809-t004:** Multivariate tests: effects of math anxiety (MA), gender (sex), and subject preference on performance on the math task in primary school children.

Effect		Value	F	Hypothesis df	Error df	Sig.	Partial Eta Squared
MA	Wilks’ Lambda	0.910	5.965	2.000	121.000	0.003	0.090
Sex	Wilks’ Lambda	0.913	5.797	2.000	121.000	0.004	0.087
Subject Preference	Wilks’ Lambda	0.895	3.433	4.000	242.000	0.009	0.054

Design: intercept + MA + sex + subject preference.

**Table 5 behavsci-14-00809-t005:** Tests of between-subjects effects of math anxiety (MA), gender (sex), and subject preference on percentage scored (Math_PT) and completion times (Math_CT) on the math task.

Source	Dependent Variable	Type III Sum of Squares	df	Mean Square	F	Sig.	Partial Eta Squared
MA	Math_PT	3884.253	1	3884.253	11.856	<0.001	0.089
Sex	Math_PT	2140.623	1	2140.623	6.534	0.012	0.051
	Math_CT	108.920	1	108.920	4.437	0.037	0.035
Subject_Preference	Math_CT	291.391	2	145.696	5.936	0.003	0.089

**Table 6 behavsci-14-00809-t006:** Multivariate tests: gender-specific effects of emotional reactivity measured by GSR (ZGSR_Math), math anxiety (MA), and subject preference and performance on the math task.

Sex	Effect		Value	F	Hypothesis df	Error df	Sig.	Partial Eta Squared
Girls	ZGSR_Math	Wilks’ Lambda	0.840	5.532	2.000	58.000	0.006	0.160
	MA	Wilks’ Lambda	0.170	5.929	2.000	58.000	0.005	0.170
	Subject_Preference	Wilks’ Lambda	0.799	3.450	4.000	116.000	0.011	0.106

Design: intercept + ZGSR_Math + MA + Subject_Preference.

**Table 7 behavsci-14-00809-t007:** Tests of between-subjects gender-specific effects of math anxiety (MA), emotional reactivity measured by GSR (ZGSR_Math), and subject preferences on percentage scored (Math_PT) and completion times (Math_CT) on the math task.

Sex	Source	Dependent Variable	Type III Sum of Squares	df	Mean Square	F	Sig.	Partial Eta Squared
Girls	ZGSR_Math	Math_CT	158.365	1	158.365	8.480	0.005	0.126
	MA	Math_PT	3394.999	1	3394.999	10.223	0.002	0.148
	Subject_Preference	Math_CT	202.922	2	101.461	5.433	0.007	0.156

**Table 8 behavsci-14-00809-t008:** Multivariate tests: effects of gender (sex) on performance on the mental rotation task.

Effect		Value	F	Hypothesis df	Error df	Sig.	Partial Eta Squared
Sex	Wilks’ Lambda	0.942	3.731	2.000	121.000	0.027	0.058

Design: intercept + ZGSR_MRT + MA + sex.

**Table 9 behavsci-14-00809-t009:** Tests of between-subjects effects of emotional reactivity measured by GSR (ZGSR_MRT), math anxiety (MA), and gender (sex) on accuracy (Sqr_Acc) and response times (Sqr_RT) on the mental rotation task.

Source	Dependent Variable	Type III Sum of Squares	df	Mean Square	F	Sig.	Partial Eta Squared
ZGSR_MRT	Sqr_RT	1.929	1	1.929	5.433	0.021	0.043
MA	Sqr_Acc	0.133	1	0.133	5.263	0.023	0.041
Sex	Sqr_RT	2.527	1	2.527	7.120	0.009	0.055

## Data Availability

The data presented in this study are available on request from the corresponding author. The data are not publicly available due to ethical and privacy reasons stated in the parental consent information form.

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
