# Peer review of "“It’s Different for Girls!” The Role of Anxiety, Physiological Arousal, and Subject Preferences in Primary School Children’s Math and Mental Rotation Performance"

_behavsci, 2024, doi:10.3390/bs14090809_

Round 1

Reviewer 1 Report

Comments and Suggestions for Authors

The introduction is too long and covers multiple topics. Sub-titles should be included to guide the reader. Additionally, the research gap and the importance of the study should be documented more consistently. For instance, clarify the contribution of the first research question in relation to the existing literature.

It would be beneficial to examine examples from mathematics tasks at different levels.

Clearly specify the analysis used for each research question. The same applies to the Analysis section. Rather than presenting different types of analysis, it would be better to organize the results section according to the research questions, instead of by the type of analysis.

Elaborate on the educational implications, especially as the study's results do not align with previous research findings.

Author Response

Dear Reviewer,

Many thanks for your patience and very helpful suggestions for improving my manuscript.

As suggested, I have edited the introduction to reduce it somewhat and have added subtitles. Additionally, I have added statements indicating how the study and the research questions address gaps in the exisiting literature.

I agree completely that an analysis of the effects of variables such as gender on performance on the more difficult items in the maths task would provide interesting insight as to whether the differences persist when complexity of items increases. Unfortunately. a limitation of the maths task is that it was not digitised so that a thorough examination of the effects of the independent variables on performance as a whole on the difficult items was not possible. We did not record individual response times on any of the maths items. Therefore, I undertook an analysis of gender differences and the effects of gender and subject preferences while controlling for maths and spatial anxety and physiological arousal on scores on a selection of the difficult maths items only. These analyses yielded no significant gender differnces and only marginal effects on scores. This therefore warrants further investigation and would be best done with a computerized maths task which records response times for individual items. I have mentioned this as a prompt for future research. I had already added the paper-and-pencil maths task as a limitation in that section.

As suggested, I have also added a separate section to the discussion with implications for education, leaving the broader societal implications in the conclusion.

I hope the revised manuscript shows improvement based on your suggestions and those of other reviewers.

Many thanks again.

Kind regards,

The Authors.

Reviewer 2 Report

Comments and Suggestions for Authors

Thank you for the opportunity to review your manuscript. The authors begin with an introduction to related literature in the field and identify a lack of research in the area on page 4, thus identifying the purpose for the research and potential impact of this work for the field. Research questions the authors identified for the study are included on page 5. The authors do a nice job of describing the number of student participants and gender of students. I appreciated the detailed description of the skin conductance. I appreciated Figure 1 to understand more about the MRT items described in the manuscript. The provide a cohesive discussion section that goes over the data analysis for each of the measures and areas. Connections are also made with literature in the discussion section to identify alignment and differences with findings from related literature. 

Recommendations:

1.        Materials and Methods (p. 5, line 255): The authors note that the population is diverse based on culture and socioeconomic background. I would recommend including information about the participants, if possible, to further support this claim. If you do not have this information, data on the student populations of the 5 schools would be another way to support this claim. 

2.        Table 1 (p. 6, Line 261): I would change the class section of the table to more clearly represent the grade levels, such as Grade 2,  2nd grade, or Second Grade. It is a little confusing how it is currently.

3.        2.1 Skin Conductance: 

a.        Indent the start of the paragraph on page 6, line 270. 

4.        Limitations       

a.        An additional limitation is that a member of the school faculty was not present during the assessment. The authors indicate that students completed the assessments in a separate room with two members of the research team. While staff was nearby, the school staff was not present in the same room. I know that this would not be allowed by IRB office for assessments with young children. This can also lead to increased anxiety when being separated from a familiar adult.

Author Response

Dear Reviewer,

Many thanks for your kind comments and useful suggestions for improving the manuscript. It is much appreciated. I have made the changes now and you will be able to see them in the revised version. Regarding comment 4, I have added this to the manuscript (procedure section) and the limitations, although it was not an issue impeding ethical approval through our university and the schools could not provide individual supervision for each group testing. However, I understand that some children may have had elevated levels of anxiety being tested by indivuals who were not members of the school staff.

The revised manuscript is uploaded with amendments based on your suggestions and those of other reviewers.

KInd regards

The Authors